# Rejection of immunogenic tumor clones is limited by clonal fraction

Ron S Gejman[1,2,3†], Aaron Y Chang[2,3†], Heather F Jones[2,3], Krysta DiKun[2,3], Abraham Ari Hakimi[4,5], Andrea Schietinger[2,6], David A Scheinberg[2,3]*

[1]Tri-Institutional MD-PhD Program, Memorial Sloan-Kettering Cancer Center, Rockefeller University, Weill Cornell Medical College, New York, United States; [2]Weill Cornell Medicine, New York, United States; [3]Molecular Pharmacology Program, Memorial Sloan Kettering Cancer Center, New York, United States; [4]Department of Surgery, Memorial Sloan Kettering Cancer Center, New York, United States; [5]Immunogenomics and Precision Oncology Platform, Memorial Sloan Kettering Cancer Center, New York, United States; [6]Immunology Program, Memorial Sloan Kettering Cancer Center, New York, United States

**Abstract** Tumors often co-exist with T cells that recognize somatically mutated peptides presented by cancer cells on major histocompatibility complex I (MHC-I). However, it is unknown why the immune system fails to eliminate immune-recognizable neoplasms before they manifest as frank disease. To understand the determinants of MHC-I peptide immunogenicity in nascent tumors, we tested the ability of thousands of MHC-I ligands to cause tumor subclone rejection in immunocompetent mice by use of a new 'PresentER' antigen presentation platform. Surprisingly, we show that immunogenic tumor antigens do not lead to immune-mediated cell rejection when the fraction of cells bearing each antigen ('clonal fraction') is low. Moreover, the clonal fraction necessary to lead to rejection of immunogenic tumor subclones depends on the antigen. These data indicate that tumor neoantigen heterogeneity has an underappreciated impact on immune elimination of cancer cells and has implications for the design of immunotherapeutics such as cancer vaccines.

DOI: https://doi.org/10.7554/eLife.41090.001

*For correspondence:
scheinbd@mskcc.org

†These authors contributed equally to this work

## Introduction

Human cancers bear uniquely distinguishable features on the surface of their cells in the form of neo-antigens, which are peptides derived from mutated, foreign or oncofetal proteins that are presented in complex with major histocompatibility complex I (MHC-I) molecules. These short, 8–11 amino acid fragments specifically mark cancer cells and activate potent immune responses that can lead to effective anti-cancer therapy (*Tran et al., 2014*; *Rosenberg and Restifo, 2015*; *Zacharakis et al., 2018*). Yet, the existence of T cells that recognize neoantigens is often not sufficient to eliminate tumors. Karl Hellström first described the coexistence of tumor-specific lymphocytes together with cancer cells in human solid tumors as a paradox 50 years ago (*Hellström et al., 1968*). In mice, an analogously enigmatic observation has been made that sporadic tumors occurring in aged or carcinogen-treated mice induce strong T cell responses only when transferred into new hosts, thereby preventing engraftment (*Heike et al., 1994*; *Dubey et al., 1997*; *Shankaran et al., 2001*). Thus, tumor-specific T cells can lead to tumor rejection in a new host or when used as a cancer therapy, but somehow immune surveillance is evaded during early tumorigenesis in the host that originally developed an immunogenic tumor. The increased rate of tumor formation in immunocompromised individuals has led to the hypothesis that the immune system can and does eliminate some

**eLife digest** T cells are specialized agents of the immune system that can detect and attack tumors. They spot their target by identifying small pieces of proteins – or antigens – at the surface of diseased cells. In particular, they can recognize the new and abnormal antigens that a cancer cell often displays. Yet, cells that become cancerous and start displaying suspicious antigens can manage to escape T cells and grow into full tumors. Why does the immune system not recognize and kill these early cancers before they get out of control?

One possibility is that T cells do not identify certain antigens carried by cancer cells. To test this, researchers have conducted experiments where they inject a mouse with cancer cells that display a single new antigen. If the animal develops a tumor, then this antigen does not trigger an immune response. However, this method is slow and laborious, because only one antigen can be tested at the time. Instead, Gejman, Chang et al. developed a new technique, PresentER, where a rodent gets injected with a mix of millions cancer cells that each displays a different antigen. This way, many thousands of new antigens can be studied in one go. The tumors are left to grow for several weeks before they are removed and analyzed to see which cells survived and which have been killed by the immune system.

Unexpectedly, the nature of the antigen did not make a big difference. Instead, cancer cells with new antigens could go undetected if they were rare and made up only a small proportion of all the different cancer cells in a tumor. However, the immune system would eliminate the exact same cancer cells when they were the major component of a cancerous lump.

Future research now has to explore exactly how rare cancer cells can hide amongst other cells, and remain invisible to the body. Armed with this knowledge, it might be possible to improve cancer therapy by prompting the immune system to target these emerging threats earlier.

DOI: https://doi.org/10.7554/eLife.41090.002

tumors, particularly virally induced tumors, before they become clinically apparent (*Schulz, 2009*). We hypothesized that if the immune system can eliminate some early tumors, but not others, perhaps it is because some antigens are more potent at inducing effective T cell responses during early tumorigenesis. Identification and characterization of neoantigens that can induce an effective immune response and clear cancer cells is critical to understanding why and how immunogenic tumors develop in immunocompetent hosts.

Immunogenic peptides have been discovered in animal studies by injection of thousands or millions of tumor cells bearing neoantigens into animals and observing tumor rejection. However, the robust immune activation and tumor rejection in these cases is not analogous to the events of early tumorigenesis in humans, when the number of transformed cells is miniscule. Thus, even though some tumors are immunogenic, it is not clear why the host cannot eliminate them when the tumors first arise. If the biochemical features of neoantigens that lead to effective T cell responses were known, it might be possible to identify which tumors bear immunogenic antigens. Some reports have linked peptide immunogenicity to the biochemical characteristics of amino acid residues at certain positions along the MHC-I ligand (*Calis et al., 2013*), while others have focused on the difference between the affinity of a wild-type ligand and a mutated ligand (*Duan et al., 2014*). However, the absence of a large, unbiased data set of known immunogenic and non-immunogenic peptides has stymied the validation of these approaches. Indeed, most known immunogenic antigens are derived from viral proteins; few mutationally-derived neoantigens are confirmed as bonafide immunogenic peptides in mice or humans. Here, we have developed a novel genetic method to express libraries of precisely defined MHC-I ligands in mammalian tumor cells and have used this method to ask questions about MHC peptide immunogenicity in immunocompetent animals during early tumorigenesis.

Using libraries of genetically encoded MHC-I ligands, we tracked the dynamic growth and depletion of thousands of tumor subclones in vivo and noted a striking failure of cancer immunosurveillance that is potentially analogous to the failure of immune surveillance in humans during early tumorigenesis. We demonstrate for the first time that the ability of the naive immune system to surveille a nascent tumor and reject immunogenic subclones is limited by the fraction of cells expressing

each unique antigen. Furthermore, we show that these rejection thresholds vary among antigens. Our data are consistent with the observation in humans that patients whose tumors have high numbers of subclones—and thus more subclonal neoantigens—have increased levels of relapse and worse survival than patients with more homogenous tumors (*McGranahan et al., 2016*; *Reuben et al., 2017*; *Turajlic et al., 2018a*). Thus, our data provide an antigen-specific rationale for the impact that tumor heterogeneity has on survival of human patients. According to our findings, antigen-specific immune effects are limited during early tumorigenesis, which has implications for the emergence and outgrowth of immunogenic tumors.

## Results

### PresentER expressing cells recapitulate known T cell immunogenicity

We have developed a reductionist method for encoding diverse peptide/MHC (pMHC) ligands in mammalian cells (termed 'PresentER'), which was purposefully designed to enable us to ask questions about the immunogenicity of individual peptide epitopes presented by cancer cells at relatively physiologic levels. We used this method to perform high-throughput, pooled screening of MHC-I ligand immunogenicity in wild-type mice (*Figure 1A*). The PresentER antigen minigene is comprised of an endoplasmic reticulum (ER) signal sequence followed by a short peptide/epitope. Expression of the peptide and its display on MHC-I does not require proteasomal degradation or peptide processing, thus enabling precise definition of the exact epitope displayed to the immune system. As previously described (*Gejman, 2018a*), Transporter associated with antigen presentation (Tap) deficient cell lines expressing PresentER antigen minigenes lead to surface presentation of the encoded MHC-I peptide, detectable by multiple modalities, including fluorescently labeled antibodies directed to specific MHC-I ligands, mass spectrometry based immunopeptidomics and antigen-specific T cell reactivity. To demonstrate the applicability of PresentER antigen minigenes to study MHC-I ligand immunogenicity, we first asked if cancer cells encoding known immunogenic (mouse Tyrp1/gp75 TAYRYHLL W223A,H224Y (*Dyall et al., 1998*); mouse Ddx5/p68 SNFVFAGI S551F (*Dubey et al., 1997*); synthetic SIYRYYGL (*Udaka et al., 1996*); synthetic VTFVFAGL (*Dubey et al., 1997*); chicken ovalbumin SIINFEKL) or non-immunogenic (mouse Serpinf1/Pedf MSIIFFLPL (*Wang et al., 2006*); scrambled chicken ovalbumin FEKIILSN; mouse Ndufa4/dEV8 EQYKFYSV (*Holler et al., 2003*); mouse Ddx5/p68 SNFVSAGI (*Dubey et al., 1997*); mouse Tyrp1/gp75 TWHRYHLL (*Dyall et al., 1998*); Mouse Trp2 SVYDFFVWL) MHC-I ligands would be rejected by wild-type (WT) animals. The C57BL/6 syngeneic, Tap deficient mouse cell line RMA/S was transduced with these antigen minigenes and we first checked that the antigens were expressed at physiologically normal levels on the cell surface. Previously we, and others, had shown that endogenously expressed MHC-I tumor antigens are presented on the cell surface at several hundred to several thousand sites per cell (*Dao et al., 2013*; *Sergeeva et al., 2011*; *Mathias et al., 2017*). Using a radiolabeled TCR mimic antibody (25-D1.16) reactive with SIINFEKL/H-2Kb, we demonstrated that there were on average 3500 binding sites per RMA/S cell expressing PresentER-SIINFEKL, compared with ~90 sites on cells expressing PresentER-MSIIFFLPL, suggesting that the PresentER system allows display of the epitopes in a relatively physiologic range. We injected $5 \times 10^6$ cells subcutaneously into WT C57BL6/N mice. Tumors expressing known non-immunogenic peptides grew, while tumors expressing known immunogenic peptides were rejected in some or most animals (*Figure 1B*). Tap deficient cells were used because peptides derived from proteins that are not directed to the ER (e.g. eGFP) are not presented on MHC as these cells lack the ability to transport peptide from the cytoplasm into the endoplasmic reticulum.

At 7 days after tumor injection, T cells specific for the chicken ovalbumin peptide SIINFEKL could be detected in tumor draining lymph nodes of animals injected with PresentER-SIINFEKL expressing cells (*Figure 1C*). Haemotoxylin and Eosin (H and E) staining of regressing PresentER-SIINFEKL tumors showed lymphocytic infiltration with hyalin rich fibrin deposits, indicating cell death. By contrast, tumors with PresentER-FEKIILSN (a non-MHC-I binding peptide) were well-vascularized, highly cellular and with trace lymphocytic infiltration (*Figure 1D*). In order to verify that the mechanism of tumor rejection was indeed T cell dependent, we injected Rag[-/-] animals with SIINFEKL or FEKIILSN positive tumors and confirmed that SIINFEKL tumors were not rejected (*Figure 1E*). Taken together,

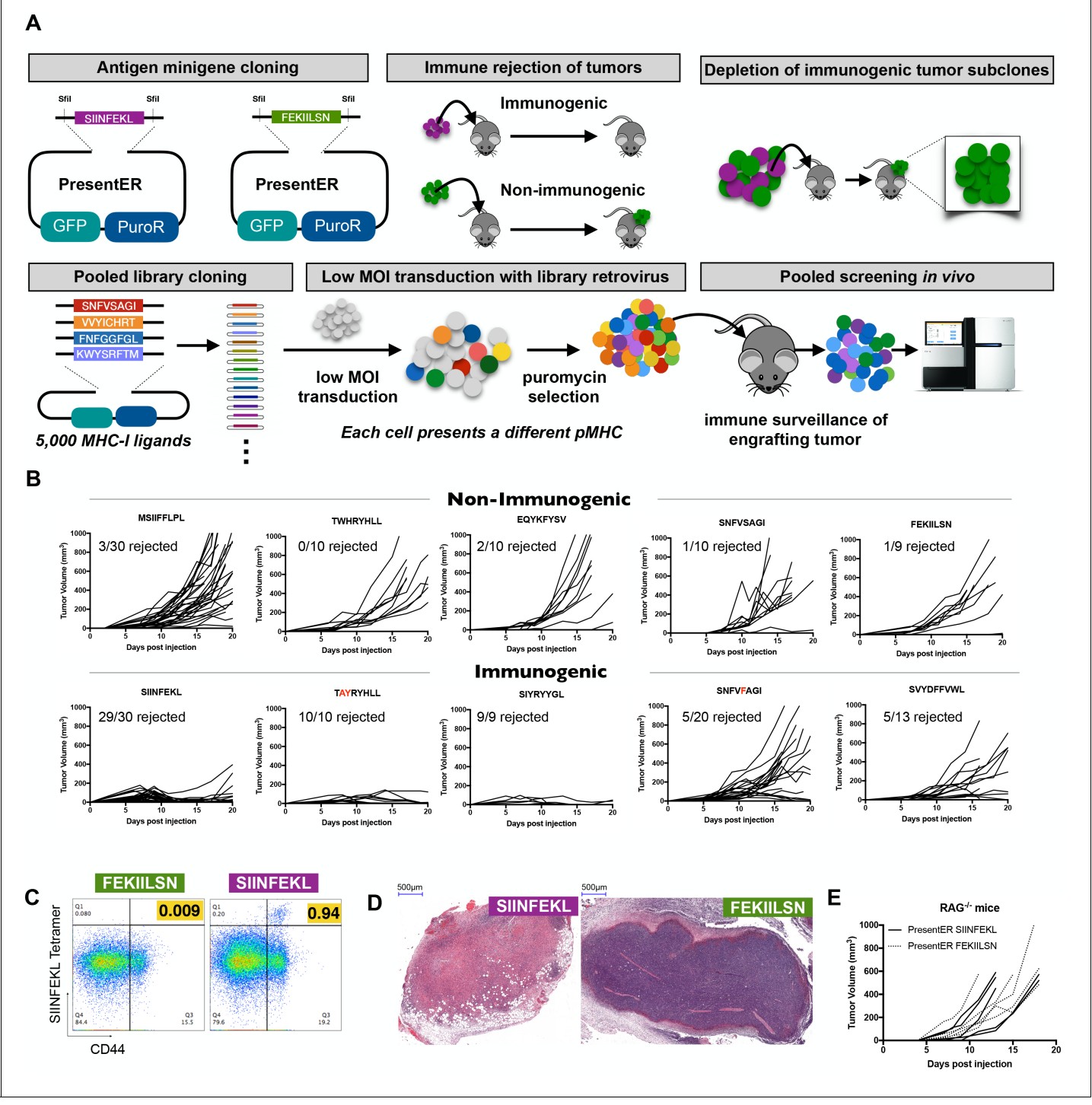

**Figure 1.** Cells expressing PresentER minigenes recapitulate the known immunogenicity of encoded antigens. (**A**) A schematic of the cloning strategy of PresentER antigen minigenes along with the experiments performed in vivo in this manuscript. (**B**) $5 \times 10^6$ RMA/S cells expressing a PresentER minigene were injected subcutaneously into C57BL6/N mice and tumor size was monitored by caliper measurements. Top row: WT or non-binding peptide minigenes. Bottom row: mutated foreign peptide minigenes. All plots are compilations of several experiments. (**C**) Mice were injected with $5 \times 10^6$ RMA/S cells expressing PresentER-SIINFEKL or PresentER-FEKIILSN. Tumor draining lymph nodes were harvested 7 days later and stained with a SIINFEKL/H-2Kb tetramer and anti-CD44. (**D**) H and E staining of tumors expressing PresentER-SIINFEKL or PresentER-FEKIILSN. (**E**) RMA/S PresentER-SIINFEKL and PresentER-FEKIILSN cells were injected into Rag$^{-/-}$ mice and tumor growth was monitored.
DOI: https://doi.org/10.7554/eLife.41090.003

these results indicate that RMA/S tumors expressing PresentER antigen minigenes can recapitulate the known immunogenicity of individual mouse MHC-I ligands in a T cell dependent manner.

## Tumors expressing PresentER antigens do not cause abscopal rejection, but do cause subclone fraction-dependent bystander killing

To study immunogenicity in vivo at high throughput and complexity in this model, we first wanted to understand if immune responses directed at immunogenic antigens lead to rejection of cells presenting non-immunogenic antigens. If so, the ability to study immunogenicity in a pooled in vivo setting might be compromised. Mice were injected with pairs of tumors expressing an immunogenic and a non-immunogenic antigen minigenes (one minigene-expressing tumor on each flank): SIINFEKL/ FEKIILSN, SIYRYYGL/EQYKFYSV and TAYRYHLL/TWHRYHLL. The immunogenic SIINFEKL, SIYRYYGL and TAYRYHLL expressing tumors were rejected, but the non-immunogenic FEKIILSN, EQYKFYSV and TWHRYHLL tumors found on the contralateral flank were not (*Figure 2A*). This suggests that tumor rejection is local and that an effective immune response to an immunogenic tumor does not affect the growth of a non-immunogenic tumor.

Next, to test if the immune system could identify and kill immunogenic subclones within a largely non-immunogenic tumor, we cultured cells with varying ratios of immunogenic (SIINFEKL) and non-immunogenic (FEKIILSN) cells. These cells were injected subcutaneously into mice and tumor size was monitored. In tumors where PresentER-SIINFEKL cells were greater than 25% of a tumor, tumors were smaller and tumor rejection occurred more frequently, especially when tumors were comprised of ≥50% immunogenic cells (*Figure 2B*). Next, we wanted to test if immunogenic subclones within a largely non-immunogenic tumor could be eliminated. We cloned mCherry into the PresentER vector and mixed PresentER-SIINFEKL (mCherry) with non-immunogenic PresentER-MSIIFFLPL (eGFP) cells at varying ratios and injected them into congenically marked CD45.1 mice. On day 16, the tumors were harvested and flow cytometry was performed to identify which (CD45.2 positive) tumor cells remained. Remarkably, within a non-immunogenic (eGFP labeled) tumor, immunogenic (mCherry labeled) sub-populations were eliminated (*Figure 2C*). Thus, in this model, the immune system is capable of recognizing and selectively depleting immunogenic tumor subclones within the context of a largely non-immunogenic tumor.

## Library screen in vivo reveals the limitations of the immune system in eliminating immunogenic tumor subclones

We hypothesized that a pooled screen in vivo might reveal the determinants of immunogenic MHC-I ligands if a tumor bearing a library of MHC-I antigen were depleted of cells bearing immunogenic peptide-MHC while tumor cells bearing non-immunogenic pMHC were spared (*Figure 3A*). Using the PresentER system, such a screen could be done on large scale and identify hundreds or thousands of immunogenic antigens at once, in contrast to identifying immunogenic antigens one-by-one. Immunoediting in vivo leading to loss of tumor clones with immunogenic neoantigen-encoding mutations has previously been observed in syngeneic mouse tumor models (*Matsushita et al., 2012*; *DuPage et al., 2012*; *DuPage et al., 2011*), suggesting that this approach might be viable.

We searched the mouse proteome in silico and randomly selected 5,000 8-mer peptides that were predicted by NetMHCPan to bind to the B6 mouse MHC-I allele H-2Kb. We also designed a library of mutated peptides by selecting single amino acid substituents of each peptide in the wild-type peptide library that did not eliminate MHC-I binding (*Figure 3B*). The MHC-I affinities and properties of the mutated peptides are further described in *Figure 3—figure supplement 1* and *Supplementary file 2*. On average, the peptides in the mutated library have slightly higher affinity for MHC-I than the peptides in the wild type library, but the plurality of mutated peptides are within 100 nM of their non-mutated counterparts (*Figure 3—figure supplement 1A–B*). The residues that are changed from the wild-type to the mutant library tend to be at positions 4, 6 and 7 (*Figure 3—figure supplement 1C*). The majority of mutations are isomorphic (polar >polar, hydrophobic >hydrophobic and charged >charged), but ~⅓ of peptides feature a hydrophobic >other substitution (*Figure 3—figure supplement 1D*). Several known immunogenic and known non-immunogenic control antigen minigenes were included in each library, including some that were described in *Figure 1*. The libraries of wild type and mutant libraries were separately cloned and introduced into RMA/S cells by transduction at multiplicity of infection <0.3, thereby

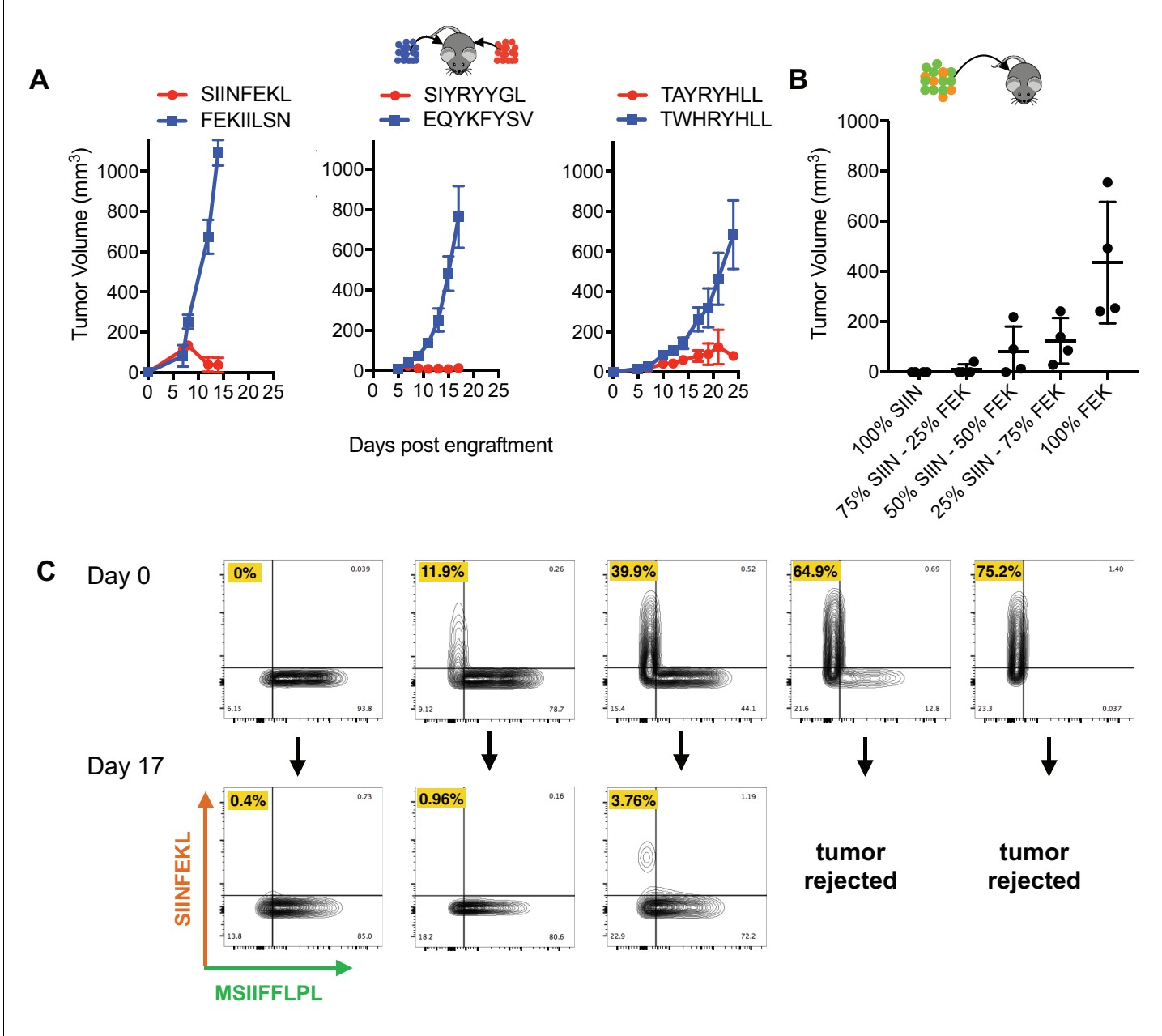

**Figure 2.** No abscopal effect and limited bystander killing of growing tumors in the RMA/S antigen minigene tumor model.  (A) Mice were injected simultaneously with $5 \times 10^6$ RMA/S cells expressing an immunogenic PresentER minigene on one flank and a non-immunogenic minigene on the contralateral flank: SIINFEKL/FEKIILSN (left; n = 3), SIYRYYGL/EQYKFYSV (middle; n = 5) and TAYRYHLL/TWHRYHLL (right; n = 5). Tumor growth curves are shown. (B) Mixtures of $5 \times 10^6$ RMA/S PresentER-SIINFEKL and PresentER-FEKIILSN injected into wild type mice (n = 4 per condition). Tumor sizes at day 15 are presented because some animals had to be sacrificed on day 17. (C) CD45.1+ mice were injected with 5 mixtures of PresentER-SIINFEKL (mCherry) and PresentER-MSIIFFLPL (eGFP) cells at several ratios. The top row shows the percentage of SIINFEKL and MSIIFFLPL cells at time 0. Tumors were harvested at day 17, enzymatically disaggregated and the percentage of CD45.2+ eGFP+ and CD45.2 mCherry+ cells were quantified (bottom row). The percentage of PresentER-SIINFEKL cells in each pretumor and tumor sample is highlighted in yellow. The two tumors with the highest percentage of PresentER-SIINFEKL cells were complete rejected and no tumor cells could be recovered.
DOI: https://doi.org/10.7554/eLife.41090.004

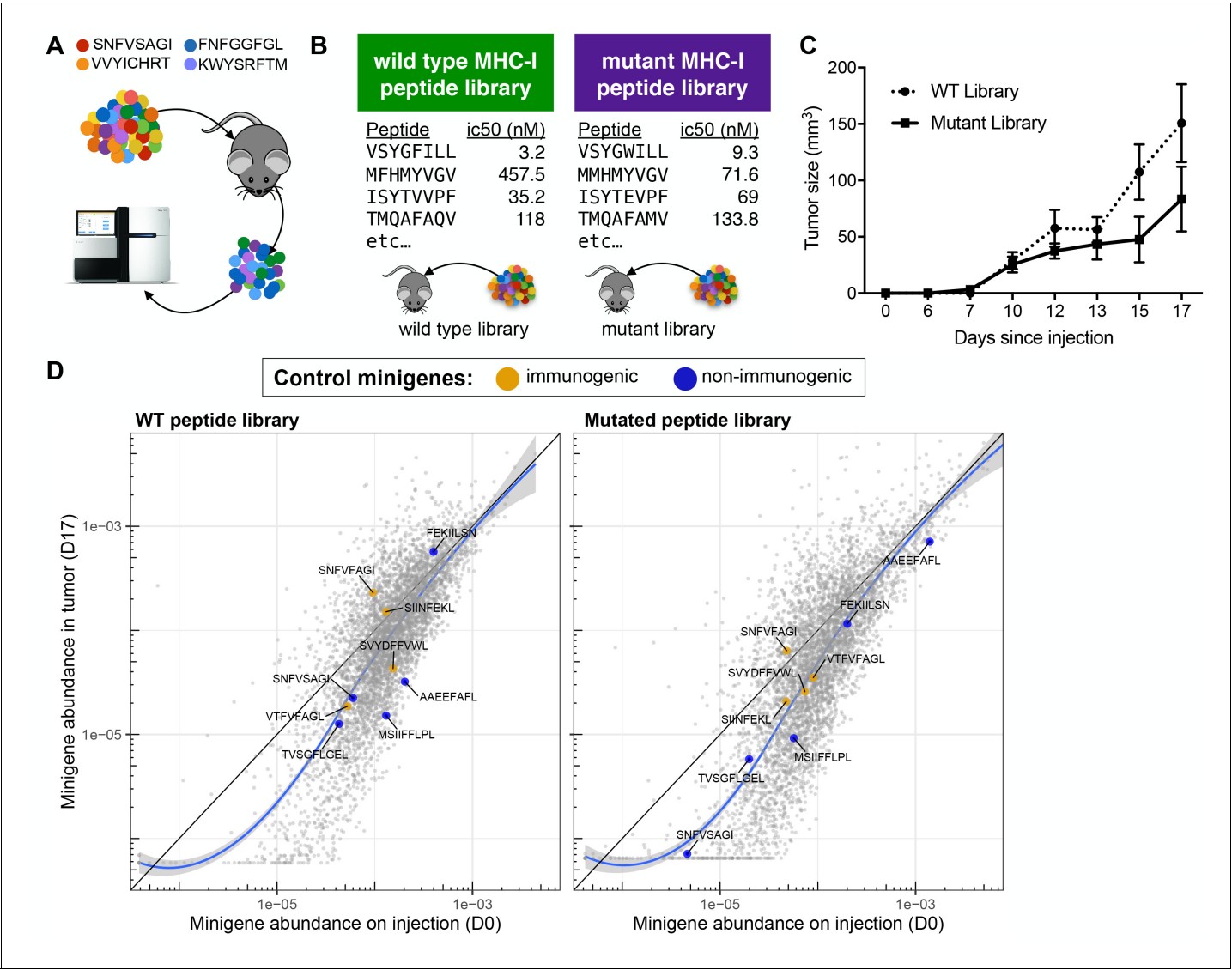

**Figure 3.** A drop-out screen for MHC-I peptide immunogenicity in immunocompetent mice. (**A**) Schematic of the drop-out screen for MHC-I immunogenicity. Mice were injected with mixtures of RMA/S cells, each of which expresses a different peptide that also served as a genetic barcode for later deconvolution. (**B**) Two libraries of mouse MHC-I peptides were constructed. Left: Wild type peptides identified by searching the mouse proteome for peptides predicted to bind to MHC-I (NetMHCpan H-2Kb ic50 <500 nM). Right: single amino acid mutants of each of the wild type peptides. (**C**) C57BL6/N mice (n = 5 per group) were injected with $5 \times 10^6$ cells bearing libraries of wild type or mutant peptides and tumor growth monitored. (**D**) A scatter plot showing the average frequency of each minigene in the library before injection of the cells (x-axis) and after growth of the tumor in wild type mice (y-axis). The abundance of each minigene before injection (n = 3) is plotted on the x-axis while the abundance of each minigene after 17 days of growth in a wild type mouse is plotted on the y-axis (n = 5). Orange circles indicate positive control (immunogenic) minigenes; blue circles indicate negative control (non-immunogenic) minigenes. The straight black lines indicate x = y. LOESS (local best fit) lines are plotted in blue. The abundance of each minigene plotted is an average of all biological replicates.

DOI: https://doi.org/10.7554/eLife.41090.005

The following source data and figure supplements are available for figure 3:

**Source data 1.** Abundance of each minigene in the pretumor and tumor samples.
DOI: https://doi.org/10.7554/eLife.41090.006

**Figure supplement 1.** Characteristics of the wild-type and mutated antigen minigene libraries.
DOI: https://doi.org/10.7554/eLife.41090.007

**Figure supplement 2.** Five C57BL6/N mice were injected with $5 \times 10^6$ RMA/S cells bearing the mutated peptide library plus $1 \times 10^6$ untransduced RMA/S cells ('padded' mutant peptide library) to provide a buffer against bystander killing in the event that most of the cells in the tumor were immunogenic.
DOI: https://doi.org/10.7554/eLife.41090.008

*Figure 3 continued on next page*

*Figure 3 continued*

**Figure supplement 3.** The abundance of each minigene in each mouse tumor after 17 days of growth in vivo (y-axis) compared to the abundance of each minigene in culture (x-axis) before injection across four groups of tumors: (A) wild type, (B) mutated, (C) 'padded' mutated and (D) mixed wild type and mutated peptide libraries.
DOI: https://doi.org/10.7554/eLife.41090.009

ensuring that few cells received more than one minigene. Naive C57BL6/N mice were injected with either (a) $5 \times 10^6$ cells expressing the wild type library, (b) $5 \times 10^6$ cells expressing the mutant library, (c) $5 \times 10^6$ mutant library cells plus $1 \times 10^6$ wild type RMA/S ('padded') or (d) $5 \times 10^6$ wild type library cells mixed with $5 \times 10^6$ mutant library cells (n = 5 per group). The tumors were allowed to grow for 17 days and then harvested (*Figure 3C* and *Figure 3—figure supplement 2A*). Genomic DNA was extracted from all of the tumors, as well as RMA/S library cells frozen on the day of injection ('pretumor' samples) and the minigenes encoded by each cell were amplified by PCR and sequenced by Illumina next generation sequencing.

Comparison of minigene abundance in the tumor outgrowth with minigene abundance in the pre-tumor samples led to the surprising finding that no minigenes were robustly depleted during growth in vivo (*Figure 3D* and *Figure 3—figure supplement 2* and *Figure 3—source data 1*). This result stood in stark contrast to the experiments described above, in which mutant peptide expressing clones were efficiently depleted in immunocompetent mice. Some minigenes that were not abundant in the library (<1/10,000) at the time of injection were depleted or dropped out entirely in the tumor due simply to stochastic drop-out; however, minigenes that were abundant in the library at time of injection maintained their abundance despite in vivo growth of the tumor. Surprisingly, even positive control minigenes encoding strongly immunogenic peptides (orange points) were not depleted in this context. Analysis of minigene abundance in individual animal tumors (as opposed to the average of several tumors) yielded the same conclusion that few, if any, minigenes were reliably depleted (*Figure 3—figure supplement 3*).

The absolute number of cells at time of injection can be estimated from the relative abundance of each minigene in the pre-tumor samples. The least abundant minigenes were found at 1 per $5 \times 10^6$ cells (0.00002%) and the most abundant at ~$23 \times 10^3$ per $5 \times 10^6$ (~0.5%). The immunogenic controls SIINFEKL, VTFVFAGL and SNFVFAGI were found at 0.005–0.013% of cells before injection into mice, which correspond to between 250 and 650 cells injected out of $5 \times 10^6$. The number of cells expressing the immunogenic antigens upon injection is very low and thus analogous to the number of cells that are present during early tumor development, when immunogenic transformed cells first arise. The surprising inability of the immune system to eliminate these demonstrably immunogenic cells may shed light on the limits of immune cell activation and killing during the early stages of tumorigenesis that enable outgrowth and escape of immunogenic cancers.

## Vaccination with minigene library does not result in immunosurveillance

A possible explanation for failure of immune surveillance in our RMA/S MHC-I minigene library model might be insufficient antigen available in the growing tumor to activate an initially productive anti-tumor immune response in naive mice. If this were true, we hypothesized that T cells from antigen experienced mice might be able to detect and kill immunogenic cells in a highly heterogeneous tumor. Although vaccination with soluble peptides has been shown to generate robust T cell immunity (*Ott et al., 2017*; *Sahin et al., 2017*), this is not cost effective at scale. Alternatively, there is precedent for the idea that vaccination with a library of mutated antigens can lead to immunogenic T cell responses that lead to slower tumor growth or clearance (*Engelhorn et al., 2006*). We decided to vaccinate mice with irradiated tumor cells bearing the library of MHC-I peptides. In order to avoid confounding immunity to the RMA/S cells themselves (*Van Hall et al., 2006*), we used CRISPR/Cas9 to generate a B6-syngeic *Tap2*$^{-/-}$ MCA205 fibrosarcoma cell line. A single cell clone of *Tap2* knockout MCA205 was selected and *Tap2* knockout was validated by RT-PCR and next generation sequencing. Decreased surface MHC-I staining was expected and observed, because the Tap complex is a key chaperone of peptide/MHC-I formation (*Figure 4—figure supplement 1*).

WT B6 mice were vaccinated three times, once every 6 days, with $1 \times 10^7$ irradiated MCA205$\Delta$*Tap2* cells bearing the wild-type library minigenes (*Figure 4A*). Splenocytes and draining lymph nodes from three vaccinated and three non-vaccinated mice were harvested at day 18 after

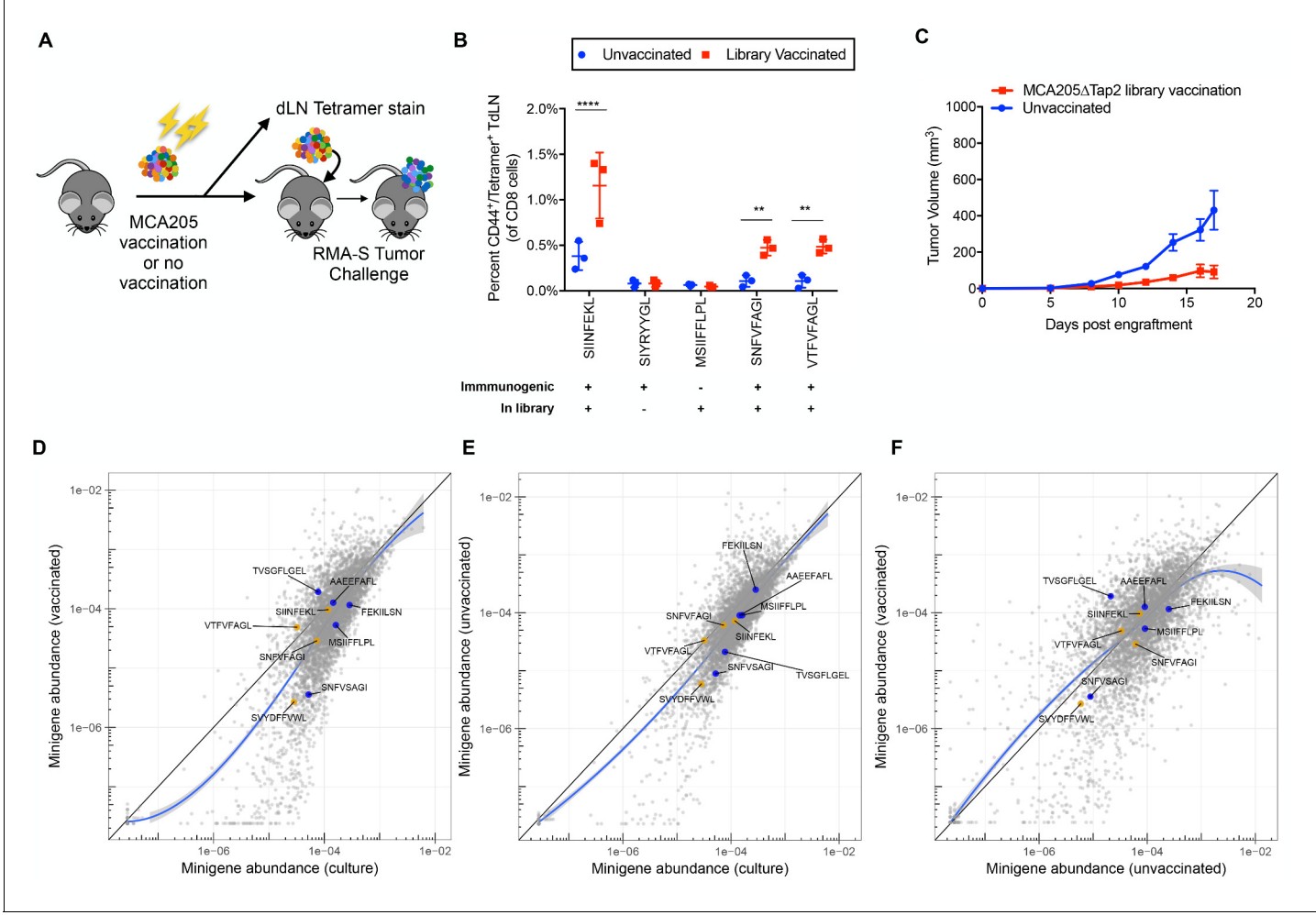

**Figure 4.** Vaccination of wild type mice with minigene library-expressing MCA205ΔTap2 cells leads to increased antigen-reactive T cells, but not increased immune surveillance (A) A schematic of the vaccinations performed on C57BL/6N mice. $10^7$ irradiated MCA205ΔTap2 cells expressing wild type library peptides were injected subcutaneously every six days (for a total of three vaccinations) into eight animals. On day 18, three mice from each group were sacrificed for tetramer analysis. Draining lymph nodes and splenocytes were stained with H-2Kb peptide tetramers. At day 18, the remaining five mice were challenged with $5 \times 10^6$ RMA-S cells expressing the library. (B) Splenocytes and draining lymph node cells from vaccinated animals were stained for CD8, CD44, and H-2Kb/peptide tetramers. Five control peptides were evaluated: four found in the library and one peptide not found in the library. The frequency of CD44/tetramer positive CD8 cells is reported. (C) Growth curves of RMA/S library tumors in in vaccinated or unvaccinated mice. (D-F) Average abundance of each minigene in cultured cells before injection into mice (x-axis) compared to minigene abundance in tumors harvested from vaccinated (n = 4; y-axis) (D) or non-vaccinated (n = 5; y-axis) (E) mice. Each circle is a minigene. Orange circles indicate positive control (immunogenic) minigenes; blue circles indicate negative control (non-immunogenic) minigenes. (F) Direct comparison of minigene abundance in tumors grown in vaccinated and non-vaccinated animals. The straight black lines indicate x = y. LOESS (local best fit) lines are plotted in blue.

DOI: https://doi.org/10.7554/eLife.41090.010

The following source data and figure supplement are available for figure 4:

**Source data 1.** Abundance of each minigene in the tumors of vaccinated and non-vaccinated animals.
DOI: https://doi.org/10.7554/eLife.41090.012

**Figure supplement 1.** MCA205ΔTap2 cell line was generated by transient transfection of MCA205 cells with a plasmid encoding Cas9 and an sgRNA directed at Tap2.
DOI: https://doi.org/10.7554/eLife.41090.011

the final vaccination and analyzed for the presence of antigen experienced T cells. Five control peptide tetramers were used, three of which are immunogenic and were present in the library (SIINFEKL, SNFVFAGI, VTFVFAGL), one which is not immunogenic but was present in the library (MSIIFFLPL) and one which is immunogenic but not found in the library (SIYRYYGL). Only the immunogenic peptides found in the library showed an increased number of CD44⁺/tetramer⁺ CD8 T cells, while the

other two peptides did not show significant changes (*Figure 4B*). Therefore, vaccination with the library yielded detectable T cell populations specific to the immunogenic peptides.

Vaccinated and non-vaccinated mice were then challenged with $5 \times 10^6$ RMA/S cells bearing the wild type peptide library. Slower tumor growth was noted in the vaccinated as compared to the non-vaccinated mice, suggesting that a vaccine-related anti-tumor effect may have occurred (*Figure 4C*). However, neither vaccinated nor unvaccinated animals showed depletion of immunogenic control minigenes in relation to the non-immunogenic control minigenes (*Figure 4D–F* and *Figure 4—source data 1*). Thus, although slower tumor growth was noted, antigen-specific immunity was not observed in response to prophylactic vaccination in the library setting, suggesting a possible response to some other, broadly expressed, cellular antigens not represented in the peptide library.

## Rejection of immunogenic subclones depends on subclone fraction in tumor

While we have demonstrated that PresentER minigenes peptides can generate effective antigen-specific immunity in bulk RMA/S tumor assays, the same response does not occur in tumors bearing libraries of MHC-I ligands. In order to test if there is a threshold level of tumor cell clonality necessary to effectively activate the immune system, CD45.1 mice were injected with mixtures of immunogenic (labeled with eGFP) and non-immunogenic (labeled with mCherry) RMA/S cells and flow cytometry was performed on reisolated tumors 17 days later (*Figure 5A*). Relative to their proportion upon engraftment, tumor cells bearing the immunogenic peptides SIINFEKL and TAYRYHLL were depleted when they comprised as little as 1% of the tumor. Below 1%, depletion of cells bearing these two minigenes could not be detected. Depletion of cells bearing the SNFVSAGI peptide could be reliably detected at 50% and in some tumors at 10%, however depletion could not be detected when the SNFVSAGI cells were found at less than 10% of the tumor (*Figure 5B*). These findings are surprising, as they indicate that immunogenic tumor subclones can persist within a tumor and that rejection or persistence is dependent on tumor cell percentage of the total tumor mass during tumorigenesis, and not immunogenicity of the cell alone.

Failure of the immune system to eliminate immunogenic subclones present at low fractional abundance could either be due to the low quantity of total antigen present in the tumor or to the low percentage of cells bearing each antigen. In order to discriminate between these two possibilities, we increased the amount of tumor injected, thus increasing the total amount of antigen the immune system sees while keeping the percentage of each antigen within the tumor the same. We grew RMA/S library tumors in Rag$^{-/-}$ mice, harvested the tumors at 20 days and retained a portion of the material for sequencing. The rest of the tumor material was transferred into the flank of WT B6 or Rag-/- mice. Each animal received approximately 1 milliliter of tumor fragments ($\sim 2.5 \times 10^8$ cells), which represents a 40–100 fold increase in cells expressing each antigen (*Figure 5C*). After 17 days, the transferred tumors were harvested and sequenced. Once again, as we observed in both naive and vaccinated mice, robust depletion of cells bearing immunogenic peptide minigenes did not occur. Overall minigene abundance was highly correlated between the tumor material transferred and the tumors harvested 17 days later in both WT and Rag-/- mice and in both the wild type and mutant libraries (*Figure 5D* and *Figure 5—source data 1*). These results reveal that it is the percentage of tumor cells bearing each antigen within a tumor and not the total quantity of antigen that determines if an effective immune response occurs.

## Discussion

The immune system is capable of recognizing cancer cells as foreign based on the presentation of altered or unusual MHC-I ligands on the surface of tumor cells—a phenomenon that has been leveraged for cancer therapies such as immune checkpoint blockade and adoptive cell transfer (*Tran et al., 2014*; *Rosenberg and Restifo, 2015*; *Zacharakis et al., 2018*). Despite this, the immune system fails to clear immunogenic, clinically detected tumors on its own, as has been paradoxically noted for decades (*Hellström et al., 1968*). It is not clear at what point in tumorigenesis the immune system begins to initiate (ineffectively) a response to immunogenic MHC-I ligands presented on cancer cells. There is epidemiologic evidence that immunocompromised patients develop more tumors, suggesting that the immune system may prevent some tumors from ever manifesting

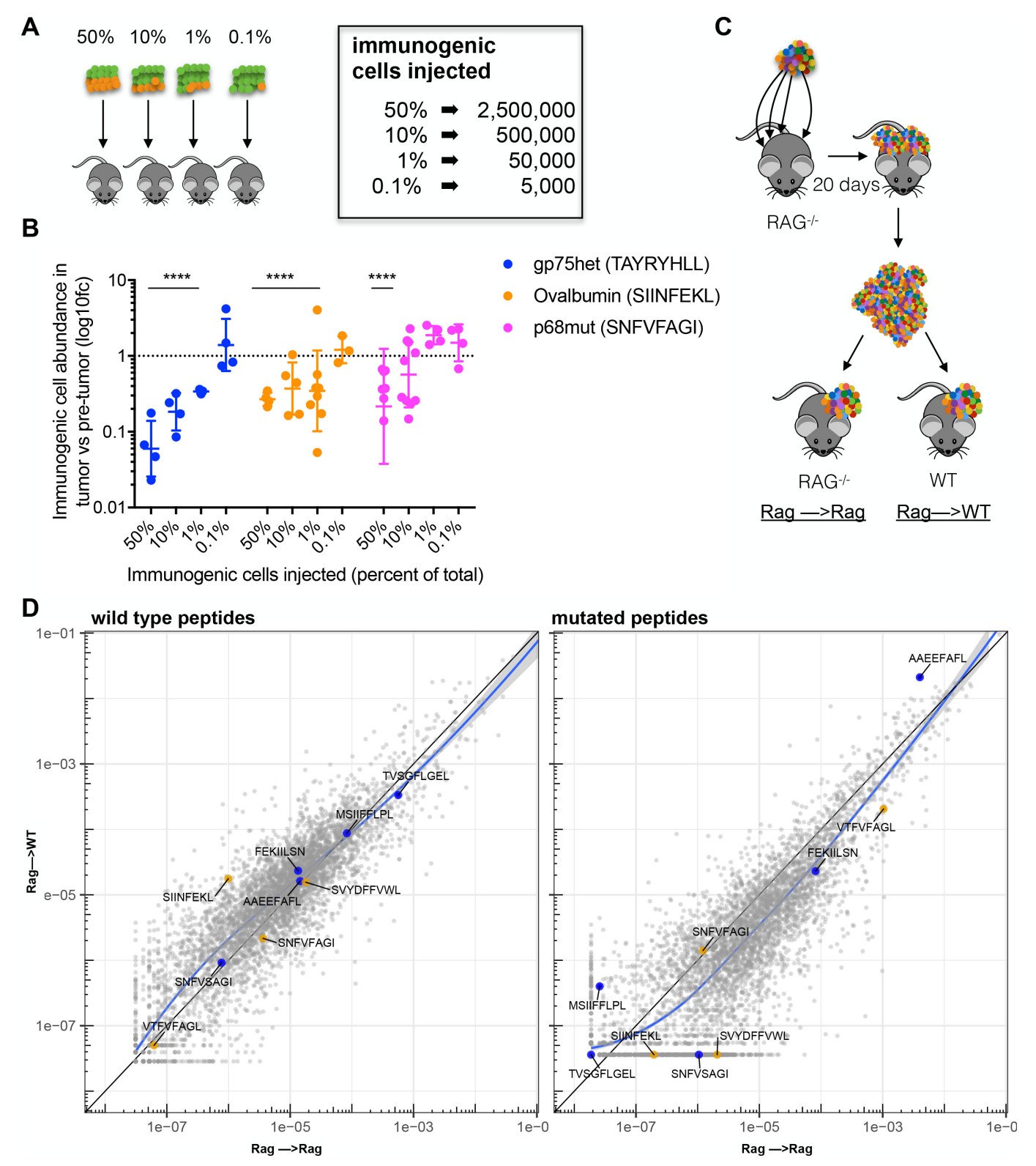

**Figure 5.** Immunosurveillance fails when tumor subclone frequency is low. (**A**) Schematic and (**B**) results of experiments to detect antigen-specific immune surveillance thresholds. CD45.1[+] C57BL/6N mice were injected with mixtures of immunogenic and non-immunogenic RMA/S cells. Mixtures were 50%, 90%, 99% or 99.9% non-immunogenic PresentER-MSIIFFLPL (mCherry) mixed with 50%, 10%, 1% or 0.1% cells expressing one of three different immunogenic minigenes (eGFP) noted in the inset. After 17 days in the mouse, tumors were enzymatically disaggregated, stained with CD45.2

*Figure 5 continued on next page*

*Figure 5 continued*

and analyzed by flow cytometry. t-tests with the alternative hypothesis that log10 fold change is 0 were performed for each group. Some tumors were rejected entirely, and these were excluded from the presented depletion analysis. At least three tumors per group yielded sufficient numbers of cells to be confident that depletion had occurred. All groups marked with four stars reached a p-value of <0.001. (C) In order to overcome the hypothesized lack of immune surveillance at low minigene abundance, 2 Rag$^{-/-}$ mice were injected at 4 sites with $5 \times 10^6$ RMA/S library cells per site. Tumors were harvested after 20 days, minced and pooled. Approximately 1 mm$^3$ of tumor fragments were implanted subcutaneously into the flank of either wild-type or Rag$^{-/-}$ mice and allowed to grow for 17 days. (D) The abundance of each minigene (average of 3 mice) in tumors transferred to WT mice (y-axis) compared to tumors transferred to Rag$^{-/-}$ mice (x-axis) is plotted. Each circle is a minigene. Orange circles indicate positive control (immunogenic) minigenes; blue circles indicate negative control (non-immunogenic) minigenes. The straight black lines indicate x = y. LOESS (local best fit) lines are plotted in blue.

DOI: https://doi.org/10.7554/eLife.41090.013

The following source data is available for figure 5:

**Source data 1.** Abundance of each minigene in the tumors transferred from RAG to WT or RAG to RAG animals.

DOI: https://doi.org/10.7554/eLife.41090.014

clinically. However, recent data also suggests that mutations accumulate at high rates in sun-damaged, but otherwise normal, tissue at levels comparable to cancer cells (*Martincorena et al., 2015*) and that tumors face overall little negative selective pressure (*Martincorena et al., 2017*). Indeed, immune escape by loss of MHC (*Masuda et al., 2007*; *Cabrera et al., 2000*; *Gudmundsdóttir et al., 2000*; *So et al., 2005*; *Shukla et al., 2015*), when it occurs, is a late and subclonal event (*McGranahan et al., 2017*; *Turajlic et al., 2018b*). If T cell surveillance were highly effective during early tumor development, negative selective pressure would be notable in the evolutionary trajectories of tumors and HLA loss would be expected to occur early, frequently and in a clonal manner. We interpret the data from human and animal experiments to suggest that routine T cell immunosurveillance of nascent cancer cells does occur, but is limited by unknown factors, and that recognition of tumors as foreign occurs mostly later in tumor development.

If tumor immunosurveillance is sometimes effective in the early growth of a tumor, we hypothesized that it might be due to potent neoantigens expressed by cancer cells. Discovery of the biochemical characteristics of mutationally-derived neoantigens that are immunogenic would be important to clarify why some neoantigens are tolerated by the immune system and others are not. The significance of this question is underscored by the major challenge currently facing cancer immunotherapy: the identification of patients whose tumors bear immunogenic neoantigens—and are thus likely to respond well to immune checkpoint blockade or other immunotherapies—vs those patients who will be unnecessarily exposed to potentially toxic therapies without a chance for efficacy.

Here, we have developed a reductionist approach to determine which MHC-I ligands are immunogenic in immunocompetent mice. We demonstrate that tumors are rejected when an immunogenic pMHC is expressed on all or most injected cancer cells. To our surprise, we have discovered that effective T cell responses are not mounted against highly immunogenic peptides when cells expressing these peptides are a minor fraction of the tumor, which is the case in early developing tumors in humans. Furthermore, the threshold percentage of tumor cells necessary to yield an antigen-specific immunogenic T cell response varies with the antigen. We propose that ineffective T cell responses may be a consequence of intratumoral (or intra-tissue) heterogeneity and that neoantigen clonal fraction is an important, overlooked aspect of MHC-I antigen immunogenicity.

There are many mechanisms by which immune mediated killing of immunogenic cancer cells can fail. Here we report the first evidence that tumor heterogeneity is an explicit factor leading to failure of effective immunity in a tumor and that the maximum level of heterogeneity tolerable before immune escape occurs is dependent on the antigen. Immune checkpoints or active tumor suppression do not explain the observed immune evasion because such mechanisms would be expected to suppress T cell killing irrespective of the fraction of tumor cells that expresses an immunogenic epitope. This is evidenced by robust depletion of the highly immunogenic peptides SIINFEKL and TAYRYHLL when present at $\geq$1% of the tumor. The mechanism by which T cell responses are restrained when immunogenic MHC-I antigens are present at low frequency is not yet clear. Low levels of antigen presentation may lead to ineffective cross-presentation of antigen in the tumor draining lymph node, thus limiting T cell activation (*Spiotto et al., 2002*). In animal models, low levels of

immunogenic epitopes of oncogenic drivers presented on transformed cells early during tumorigenesis (at the premalignant stage) were shown to induce a program of cellular hypo-responsiveness in tumor-specific (oncogene-specific) CD8 T cells (*Willimsky and Blankenstein, 2005*; *Schietinger et al., 2016*). After prophylactic vaccination with irradiated library cells we observed smaller tumors and increased numbers of antigen-specific T cells in splenocytes and draining lymph nodes, suggesting that some antigen-specific T cell proliferation does occur. While we demonstrated sufficient levels of the high affinity, highly immunogenic antigens on the cell surface, tumor associated epitopes of lower affinity, immunogenicity and expression levels should produce an even more pronounced deficit in T cell recognition. We speculate that the mechanism of immune escape in this model is either ineffective T cell activation or failure of activated T cells to identify and kill antigen-positive cells present at low abundance within a sea of cells displaying irrelevant antigens.

The observation that animals prophylactically vaccinated with MCA205ΔTap2 cells are partially protected from outgrowth of RMA/S tumors may be due to several possibilities. For instance, the vaccinations may have activated T cells that recognize some immunogenic peptides in the library, or T cells that recognize mutated peptides expressed by both cell types, or, most intriguingly, T cells specific to T cell epitopes associated with impaired peptide processing (TEIPPs). TEIPPs are T cell epitopes derived from wild type proteins that are not normally presented, unless antigen presentation is altered by loss of Tap, which yields a relative deficit of peptides translocated into the ER. TEIPPs in RMA/S cells can be immunogenic (*Van Hall et al., 2006*; *Doorduijn et al., 2016*). Although RMA/S and MCA205ΔTap2 are unlikely to share neoantigens derived from somatic mutations, they may share TEIPPs that contribute to protection against RMA/S tumor growth. A caveat to the vaccination experiments is that differences in the antigen presentation machinery of MCA205ΔTap2 and RMA/S cells may lead to incomplete congruence in the MHC-I peptidome of these two cell lines. These differences may lead to some peptides encoded by the PresentER library (or by the endogenous genome of the cell) to be poorly presented in one cell line but well presented in another cell line.

Although we have focused our efforts on understanding the role of CD8 T cells, other infiltrating cell types are also important for anti-tumor effects. In particular, CD4 T cells have been shown to play important roles in the CD8 T cell responses to tumors (*Borst et al., 2018*) and to target neoantigens (*Ott et al., 2017*; *Sahin et al., 2017*). However, our experimental design does not allow us to assess the role of CD4 T cells in CD8 T cell mediated anti-tumor responses. Thus, while it may be the case that the magnitude or quality of the immune response may differ when both CD8 and CD4 cells recognize a related immunogenic antigen, we are unable to study this with the tools we have developed to perform pooled library experiments because we cannot encode both a specific MHC II and MHC I ligand in a single minigene. Moreover, there is the confounding possibility that some CD4 help is coming from unrelated antigens, such as mutated proteins found in RMA/S or the exogenous proteins introduced into the cells by the PresentER vector. More broadly, the heterogeneity in actual human tumors is characterized by differential infiltration with antigen presenting cells, which can change the sensitivity of the immune system to detect immunogenic antigens present at low clonal fractions, and thus could impact our findings. Although we could not modulate the infiltrating immune cells in our experimental tumor models, we appreciate that this may influence the potency of a subpopulation of immunogenic cells. It would be interesting to understand how various levels of antigen presenting cells contribute to dynamic thresholds of T cell immunogenicity.

The PresentER system we have employed does have some potential biochemical caveats that may impact the interpretation of the data. While we have not observed alternative cleavage patterns in individual peptides we have studied (*Gejman, 2018b*), in a library setting we cannot test if every minigene is properly yielding its encoded MHC-I ligand. For instance, some encoded peptides may not bind well to MHC-I or may only bind in the presence of an MHC loading chaperone. Some peptides may lead to improper or no cleavage of the signal peptide whereas other peptides may be shortened (e.g. by ERAP1, an ER associated endopeptidase), altered or destroyed in the endoplasmic reticulum.

New therapeutic modalities such as immune checkpoint blockade have shown clinical efficacy in the treatment of human tumors, but the toxicity of the drug regimens has led to many efforts to find biomarkers that predict patient response. Tumors with high mutation burdens—which are more likely to have immunogenic neoantigens (*Van Allen et al., 2015*; *Yarchoan et al., 2017*; *Snyder et al., 2014*)—and tumors that have mismatch repair (MMR) deficiency or microsatellite

instability (MSI-H) respond well to checkpoint blockade (*Le et al., 2015*; *Le et al., 2017*). In mouse models, syngeneic tumors with MMR gene knock outs accumulate mutations over time and grow more slowly in wild type mice than do tumors without MMR deficiency. Cell lines derived by sub-cloning of MMR deficient lines—thus increasing the clonal fractions of each neoantigen—grow more slowly or are rejected entirely. In all cases, MMR inactivation in tumors leads to better responses to checkpoint blockade than the parental tumors (*Germano et al., 2017*). Moreover, survival is inversely related to tumor neoantigen clonality in human patients with lung adenocarcinoma (*McGranahan et al., 2016*; *Reuben et al., 2017*). Patients whose tumors bear high numbers of sub-clonal (or branched) neoantigens have increased levels of relapse and worse survival than those patients with more homogenous tumors (*McGranahan et al., 2016*; *Reuben et al., 2017*). Response to checkpoint blockade in lung and skin tumors is also associated with lower levels of intratumoral heterogeneity (*McGranahan et al., 2016*). The combination of these data are highly suggestive that while total neoantigen burden is important for long term survival and response to checkpoint block-ade, neoantigen clonality is an additional important factor in mediating tumor regression.

The discovery that intratumoral heterogeneity prevents effective T cell responses has implications for the development of therapeutic cancer vaccines, checkpoint blockade, adoptive T cell therapies, and studies of tumor immunogenicity. In addition, our data may provide a new understanding of mechanisms of immune surveillance and its failure that allows growth and evolution of tumors and subclones. For instance, recent data shows that subclones with very low clonal fraction yet distinct intratumoral functions may be important to tumor survival and growth (*Vinci et al., 2018*)—suggest-ing that even low abundance tumor subclones may be important targets for immunotherapy. In gen-eral, we may speculate on one mechanism for cancer escape and progression, in which small early cancers generally do not bear immunogenic epitopes derived from their limited number of driver oncogenic proteins and few passenger mutations. Then as the tumors evolve, the neoantigens appearing in subclones do not reach a clonal fraction high enough to breach their antigen-specific immunogenicity thresholds, thereby allowing escape of these otherwise immunogenic clones. This model may help to explain why sun-damaged and aged healthy tissue is replete with mutations, sometimes reaching frequencies seen in human cancers (*Martincorena et al., 2015*; *Risques and Kennedy, 2018*). Cumulatively, these findings suggest that effective T cell immunity is restrained in the context of healthy tissue and growing tumors and paint a picture of T cell immunogenicity that is poorly captured by existing models of tissue immunosurveillance.

## Materials and methods

**Key resources table**

| Reagent type (species) or resource | Designation | Source or reference | Identifiers | Additional information |
|---|---|---|---|---|
| Cell line (M musculus) | RMA/S | 10.1002/ijc. 2910470711 | RRID:CVCL_2180 | |
| Cell line (M musculus) | MCA-205 | PMID: 2303716 | MCA-205 | |
| Cell line (M musculus) | MCA205ΔTap2 | this paper | | Generated using CRISPR/Cas9 described in materials/methods |
| Recombinant DNA reagent | PresentER-SIINFEKL | this paper | | Generated using procedure described in materials/methods |
| Recombinant DNA reagent | PresentER-FEKIILSN | this paper | | Generated using procedure described in materials/methods |
| Recombinant DNA reagent | PresentER-MSIIFFLPL | this paper | | Generated using procedure described in materials/methods |

*Continued on next page*

*Continued*

| Reagent type (species) or resource | Designation | Source or reference | Identifiers | Additional information |
|---|---|---|---|---|
| Recombinant DNA reagent | PresentER-SNFVSAGI | this paper | | Generated using procedure described in materials/methods |
| Recombinant DNA reagent | PresentER-SNFVFAGI | this paper | | Generated using procedure described in materials/methods |
| Recombinant DNA reagent | PresentER-SIYRYYGL | this paper | | Generated using procedure described in materials/methods |
| Recombinant DNA reagent | PresentER-EQYKFYSV | this paper | | Generated using procedure described in materials/methods |
| Recombinant DNA reagent | PresentER-TWHRYHLL | this paper | | Generated using procedure described in materials/methods |
| Recombinant DNA reagent | PresentER-TAYRYHLL | this paper | | Generated using procedure described in materials/methods |
| Recombinant DNA reagent | LentiCRISPRv2 | Addgene | Addgene:52961 | |
| Strain, strain background (m musculus) | C57BL6/N | Envigo | Envigo:44 | |
| Strain, strain background (m musculus) | B6.SJL-Ptprca/BoyAiTac | Taconic Biosciences | Taconic:4007 F | |
| Strain, strain background (m musculus) | B6.129S6-Rag2tm1Fwa N12 | Taconic Biosciences | Taconic:RAGN12-F | |
| Antibody | anti-H-2Kb | Biolegend | Biolegend:116517 | used at 1:200 |
| Antibody | anti-SIINFEKL /H2kb | Biolegend | Biolegend:141605 | used at 1:200 |
| Antibody | anti-CD45.2 | eBioscience | eBioscience:17-0454-81 | used at 1:400 |
| Antibody | anti-CD8a | BD Pharmingen | BD Pharmingen:553030 | used at 1:200 |
| Antibody | anti-CD3 | BD Pharmingen | BD Pharmingen:553067 | used at 1:400 |
| Antibody | anti-CD44 | Biolegend | Biolegend:103012 | used at 1:400 |
| Other | H-2Kb SIINFEKL tetramer | NIH Tetramer Core | NIH Tetramer Core:K(b)/Ova.SIINFEKL | used at 10 nM |
| Other | H-2Kb SIYRYYGL tetramer | NIH Tetramer Core | NIH Tetramer Core:K(b)/SIYRYYGL | used at 10 nM |
| Other | H-2Kb MSIIFFLPL tetramer | NIH Tetramer Core | NIH Tetramer Core:custom | used at 10 nM |
| Other | H-2Kb SNFVFAGI tetramer | NIH Tetramer Core | NIH Tetramer Core:K(b)/mp68.SNFVFAGI | used at 10 nM |
| Other | H-2Kb VTFVFAGL tetramer | NIH Tetramer Core | NIH Tetramer Core:custom | used at 10 nM |

## Animal studies

6-8 week old C57BL6/N mice were purchased from Envigo or Taconic Biosciences. 6–8 week old B6. SJL-*Ptprc*$^a$/BoyAiTac (known as CD45.1 mice) and B6.129S6-Rag2$^{tm1Fwa}$ N12 Mice (known as Rag2 KO) were purchased from Taconic Biosciences. Mice were shaved before subcutaneous engraftment of the indicated number of RMA/S cells in 100 uL PBS. Tumor volumes were calculated using caliper measurements and the standard modified ellipsoid formula: tumor volume = (LxW$^2$ x 0.52 every 2–3 days. Animals were euthanized when tumor volume exceeded 2000 mm$^3$ or if ulceration was noted. Vaccination of animals was performed by subcutaneous injection of $10 \times 10^6$ irradiated (20 Gy) MCA205-ΔTap2 cells expressing libraries of minigenes.

## Flow cytometry and radioimmunoassay

For cell surface staining, cells were incubated with appropriate fluorophore-conjugated mAbs for 30 min on ice and washed twice before resuspension in the viability dye DAPI at 1 µg/mL. Flow cytometry data were collected on a LSRfortessa (BD) or an Accuri C6 (BD) and analyzed with FlowJo V10 software. The antibodies used in this study were anti-H-2Kb-APC clone AF6-88.5 (Biolegend 116517), anti-SIINFEKL/H2kb-APC clone 25-D1.16 (Biolegend 141605), anti-CD45.2-APC clone 104 (eBioscience 17-0454-81), anti-CD8a-FITC clone Ly-2 (BD Pharmingen 553030), anti-CD3-PerCP clone 145–2 C11 (BD Pharmingen 553067). The following fluorescently labeled H-2Kb tetramers were obtained through the NIH Tetramer Core Facility: SIINFEKL, SIYRYYGL, MSIIFFLPL, SNFVFAGI and VTFVFAGL. Radioimmuneassay to determine the number of SIINFEKL/H-2Kb molecules on the surface of RMA/S PresentER-SIINFEKL cells was performed as previously described (*Dao et al., 2013*; *Chang et al., 2017*).

## Generation of MCA205-ΔTap2

A guide RNA sequence targeting murine Tap2 (ATGGGGCTGTTGCGCTGAGC) was cloned into the LentiCRISPRv2 (*Sanjana et al., 2014*) plasmid (Addgene plasmid 52961), a gift from Feng Zhang (Broad Institute, Cambridge, Massachusetts, USA). MCA205 fibrosarcoma cells were transiently transfected using Lipofectamine 2000 (Thermo Fisher Scientific 11668027) following standard manufacturer's protocols. 24 hr later, successful transfectants were selected using 5 ug/mL Puromycin for 3 days before expansion and single-cell subculture. Genetic ablation of Tap2 was verified by next generation sequencing of the Tap2 loci confirming a frameshift deletion in both alleles, and RT-PCR analysis. Primers for RT-PCR analysis are listed in *Supplementary file 1*. Reduced cell-surface H-2Kb expression was also verified by flow cytometry.

## Determination of antigen-specific immunogenicity thresholds

RMA/S cells expressing PresentER antigen #1 (eGFP) were mixed with RMA/S cells expressing PresentER antigen #2 (mCherry) at defined ratios (e.g. 1:10, 1:100, 1:1000, etc) and validated by flow cytometry immediately before injection into CD45.1 mice. After 17 days, tumors were harvested, cut into small pieces and disaggregated by incubation at 37°C with Liberase TL (Sigma-Aldrich 5401020001), DNAse I (Worthington Biochemical LS002139) and ACK lysis buffer (Thermo Fisher Scientific A1049201). Single cell suspensions of tumor cells were stained with CD45.2 and DAPI and collected on the same flow cytometer, using the same settings and gates as on day 0. The number of eGFP and mCherry cells was calculated based on the same gates used on the day of injection. Non-fluorescent cells were ignored for the purposes of analysis and the percentage of eGFP and mCherry cells in the tumor were normalized to sum to 100%. The fold change in cells expressing each antigen was calculated as: (percentage of normalized eGFP cells in tumor) / (percentage of eGFP cells at D0).

## PresentER minigene cloning and transduction of RMA/S

The individual PresentER minigenes specified above were cloned into the PresentER backbone as previously described (*Tran et al., 2014*). Oligonucleotides were amplified using T3_SfiI and T7_SfiI primers (*Supplementary file 1*). HEK293T Phoenix-Ampho cells were transfected with each plasmid and, after 24 hr, viral supernatant was harvested every 12 hr. RMA/S were transduced with limiting amounts of viral supernatant at 1000xg for 2 hr at 37°C in six well non-tissue culture treated plates. PresentER minigenes vectors are available on Addgene (102942, 102943, 102944, 102945, 102946).

## Library construction in silico

The peptides included in the libraries were found in Uniprot database of canonical mouse protein sequences (UP000000589). Substrings of unique eight amino acid sequences were collected and affinity to H-2Kb was calculated using NetMHCPan v4.0. 5000 randomly selected peptides with predicted ic50 <500 nM were selected and constitute the 'wild type peptide library.' A single random amino acid substitution was made to each member of the wild type library to generate the 'mutant peptide library.' Substitutions which generated another wild type peptide were excluded.

## Library generation in vitro

Libraries of PresentER minigenes were cloned as previously described (*Tran et al., 2014*). Library metadata is provided in *Supplementary file 2*. Oligonucleotide libraries were ordered from Custom-Array and amplified with Phusion polymerase using WT and Mutant minigene library-specific primers (*Supplementary file 1*). Amplicons were digested with SfiI and passed through a MinElute column. The PresentER cassette vector was also digested with SfiI, treated with calf intestinal phosphatase and ligated to the oligonucleotides with T4 ligase. Ligation products were phenol extracted and electroporated into DH5 electrocompetent cells. Electroporated cells were plated and counted. At least 1000x fold more transformants than minigene library members were required to proceed to plasmid DNA extraction (>$5\times10^6$ colonies). The colonies were scraped off the plate and grown for 3.5 hr in TB +ampicillin at 37°C at 225 rpm. The bacteria were maxiprepped using the Qiagen maxiprep kit and library representation was checked by Illumina sequencing. Library containing retrovirus was produced by transfection of 15 cm plates of HEK293T Phoenix-AMPHO cells with library plasmid DNA. Viral supernatant fluid was collected beginning at 24 hr after transfection and continuing every 12 hr until 72 hr post transfection. Viral supernatant fluid was pooled, concentrated with Clontech Retro-X concentrator and frozen. Concentrated viral supernatant fluid titers were determined by transduction of RMA/S cells. Libraries of minigene expressing RMA/S cells were generated by transduction at an MOI <0.3 (1,000xg for 2 hr at 37°C in six well non-tissue culture treated plates) and selection with puromycin. Library expressing cells were maintained in cultures of >1000 x cells per number of minigenes in the library (i.e. at least $5 \times 10^6$).

## Genomic DNA extraction and minigene sequencing

To verify that minigene representation was not compromised during cloning, all libraries were sequenced from plasmid prior to transduction into mammalian cells. Minigenes were either directly amplified with P5 and barcoded P7 primers and sequenced using a custom, or they were amplified with a nested PCR protocol followed by Illumina library preparation and sequencing at the Integrated Genomics Operation at MSKCC (*Supplementary file 1*). Genomic DNA was extracted from cultured cells or mechanically disaggregated tumors with the Gentra Puregene kit and minigenes were amplified from genomic DNA. In order to avoid amplification bias due to high PCR cycle numbers or multiple integration events, PCR was performed with the minimum number of cycles possible. The minimum number of PCR cycles was empirically-determined for each batch of samples, but was typically 18–22 cycles when the template was genomic DNA. All steps were performed in PCR hoods or other clean environments with precautions to avoid cross contamination between samples. In order to avoid under-sampling low abundance minigenes, each sample of genomic DNA was amplified using DNA equivalent to >1,000 fold the number of minigenes in the library. Reads were mapped to the PresentER minigene libraries with Bowtie2 using default settings. Reads that did not map to the minigenes in the library were discarded. The number of reads aligning to each minigene was divided by the total number of reads aligning to the library to obtain the relative abundance of each minigene. Quality control to ensure library representation and absence of contamination were performed on every sample. All data analysis was performed in R.

## Data and materials availability

Some PresentER minigenes are available from Addgene (e.g., #102942, #102943, #102945, #102946, #102944) and others are available by request directly from the investigators. Data are available in the following repositories: DOI: 10.5281/zenodo.1310902, DOI: 10.5281/zenodo.1309836 and DOI: 10.5281/zenodo.1308909.

## Acknowledgments

We graciously acknowledge the help and support of the Integrated Genomics Operation, Flow Cytometry and Molecular Cytology core facilities at Memorial Sloan Kettering. We thank Andrew Scott for advice and help with flow cytometry staining panels. We thank Justin Mulvey for assistance with analysis of pathology slides. Many thanks to Taha Merghoub, Mathieu Gigoux, Martin Klatt, Hans Schreiber and Karen Schreiber for useful conversations

## Additional information

### Competing interests

Ron S Gejman: Memorial Sloan Kettering Cancer Center has filed for intellectual property protection for the inventions of this author in relation to the PresentER method. Aaron Y Chang: Memorial Sloan Kettering Cancer Center has filed for intellectual property protection for the inventions of this author in relation to TCR mimic antibodies. David A Scheinberg: Memorial Sloan Kettering Cancer Center has filed for intellectual property protection for the inventions of this author in relation to the PresentER method and TCR mimic antibodies. DAS is also a board member of, or consultant to, and/or owns equity in SLS, IOVA, PFE, and Eureka therapeutics that work in the immunotherapy field. The other authors declare that no competing interests exist.

### Funding

| Funder | Grant reference number | Author |
| --- | --- | --- |
| Memorial Sloan-Kettering Cancer Center | Functional Genomics Initiative | Ron S Gejman Abraham Ari Hakimi Andrea Schietinger David A Scheinberg |
| National Institutes of Health | DP2 CA225212 | Abraham Ari Hakimi |
| Josie Robertson Foundation | | Andrea Schietinger |
| National Cancer Institute | PO1 CA 55349 and CA23766 | David A Scheinberg |
| National Institute of General Medical Sciences | T32GM07739 | Ron S Gejman |
| National Cancer Institute | F30 CA200327 | Ron S Gejman |
| Memorial Sloan-Kettering Cancer Center | Core Grant P30 CA008748 | Abraham Ari Hakimi |

The funders had no role in study design, data collection and interpretation, or the decision to submit the work for publication.

### Author contributions

Ron S Gejman, Conceptualization, Resources, Data curation, Software, Formal analysis, Funding acquisition, Validation, Investigation, Visualization, Methodology, Writing—original draft, Project administration, Writing—review and editing; Aaron Y Chang, Conceptualization, Resources, Formal analysis, Validation, Investigation, Methodology, Writing—original draft, Writing—review and editing; Heather F Jones, Investigation, Writing—review and editing; Krysta DiKun, Conceptualization, Investigation, Methodology; Abraham Ari Hakimi, Conceptualization, Funding acquisition, Writing—review and editing; Andrea Schietinger, Conceptualization, Resources, Supervision, Funding acquisition, Methodology, Writing—original draft, Writing—review and editing; David A Scheinberg, Resources, Supervision, Funding acquisition, Writing—original draft, Writing—review and editing

### Author ORCIDs

Ron S Gejman http://orcid.org/0000-0001-5138-2772
Aaron Y Chang https://orcid.org/0000-0002-7770-7405
Heather F Jones http://orcid.org/0000-0002-9459-377X

Abraham Ari Hakimi ⓘD http://orcid.org/0000-0002-0930-8824
David A Scheinberg ⓘD http://orcid.org/0000-0002-4160-923X

## Ethics

Animal experimentation: Experiments involving animals were conducted with the approval of an IACUC at MSKCC (Protocol #96-11-044).

## Decision letter and Author response

Decision letter https://doi.org/10.7554/eLife.41090.025
Author response https://doi.org/10.7554/eLife.41090.026

## Additional files

### Supplementary files

• Supplementary file 1. Tables of oligonucleotide sequences used in this manuscript, including qPCR primers, cloning, amplification and Illumina sequencing oligonucleotides.
DOI: https://doi.org/10.7554/eLife.41090.015

• Supplementary file 2. Metadata corresponding to the wild type and mutant PresentER minigene libraries: peptide, gene, NetMHCPan predicted H-2Kb ic50 and type of mutation.
DOI: https://doi.org/10.7554/eLife.41090.016

• Transparent reporting form
DOI: https://doi.org/10.7554/eLife.41090.017

### Data availability

PresentER plasmids are available from Addgene (#102942, #102943, #102945, #102946, #102944). Data are available in the following repositories: DOI: 10.5281/zenodo.1310902, DOI: 10.5281/zenodo.1309836 and DOI: 10.5281/zenodo.1308909.

The following datasets were generated:

| Author(s) | Year | Dataset title | Dataset URL | Database and Identifier |
|---|---|---|---|---|
| Gejman RS, Scheinberg DA | 2018 | Outgrowth of transferred tumors expressing libraries of PresentER minigenes in immunocompetent and immunodeficient mice | http://doi.org/10.5281/zenodo.1310902 | Zenodo, 10.5281/zenodo.1310902 |
| Gejman RS, Scheinberg DA | 2018 | Outgrowth of tumors expressing libraries of PresentER minigenes in vaccinated or unvaccinated immunocompetent mice | http://doi.org/10.5281/zenodo.1309837 | Zenodo, 10.5281/zenodo.1309836 |
| Gejman RS, Scheinberg DA | 2018 | Outgrowth of tumors in immunocompetent mice expressing libraries of PresentER minigenes | http://doi.org/10.5281/zenodo.1308910 | Zenodo, 10.5281/zenodo.1308909 |

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
