## [Decision Letter]

Thank you for submitting your article "Rejection of immunogenic tumor clones is limited by clonal fraction" for consideration by *eLife*. Your article has been reviewed by two peer reviewers, and the evaluation has been overseen by a Reviewing Editor and Tadatsugu Taniguchi as the Senior Editor. The following individual involved in review of your submission has agreed to reveal his identity: K Melief (Reviewer #1).

Your careful quantitative analysis of the role of tumor heterogeneity (clonal fraction) in effective T cell elimination of cancer adds an important new level of insight into the reasons for failure of immunotherapy even when a tumor possesses adequate neoantigens for immune recognition and effector T cells specific for that antigen exist.

The reviewers have discussed the reviews with one another and the Reviewing Editor has drafted this decision to help you prepare a revised submission.

Summary:

This paper in an ingenious fashion approaches experimentally the important issue of tumor heterogeneity and immunogenicity. To that purpose the authors have used the antigen processing defective (TAP defective) RMA-S cell-line and supplied it with minigene constructs that allow the MHC class I epitope direct access to the ER for MHC class I loading, circumventing the TAP defect in RMA-S. The startling finding is then made through a number of well-designed experiments that small numbers of tumor cells, presenting a given model neo-epitope in a sea of other neo-epitopes escape being noted by the immune system, leading to tumor outgrowth. Not entirely surprisingly, for some epitopes larger fractions of tumor cells go unnoticed than for others, probably reflecting the " quality" of each epitope. While the interpretation offered by the authors in general may be appropriate, additional consideration apply, which they should discuss, because this adds to the relevance of their paper. If true, the conclusions imply a novel parameter strongly influences immunogenicity/targeting of tumor cells and could explain empirical observations on frequent development of heterogenous human tumors.

Essential revisions:

1) The authors have not reckoned with a potential presentation of epitopes by MHC class II and possible effects of CD4 help. It is well known that optimal CD8^+^ T cell immunity depends bot for effector cell induction and memory on specific CD4^+^ T cell immunity. Specific help is better than non-specific help, because the CD4^+^ T cells not only assist in the induction and programming of optimally activated CD8^+^ T cells, but also improve the intra-tumoral conditions for activity of CD8^+^ T cells. In the average neo-antigen carrying human tumors both CD4 and CD8 neo-epitopes exist (see Ott et al., 2017 and Sahin et al., 2017 papers quoted in the manuscript). It would be great to probe the rules for immunogenicity ageing in the presence or absence of specific helper epitopes being expressed and presented from the same RMA-S tumor, but I appreciate the fact that this is asking too much at the present time. Nevertheless it is quite possible that smaller fractions of tumor cells would become eliminated in the presence of robust and tumor-specific CD4^+^ help. The importance of CD4^+^ help has been discussed recently in Nature Reviews Cancer by Jannie Borst et al., 2018.

2) Tumor heterogeneity may also involve the fact that some tumors are very rich in APC, while others are not, lacking the appropriate chemokine (receptor) expression. This type of heterogeneity might heavily influence the results one might obtain with the system used by the authors. Moreover most tumors lack intrinsic danger signals to activate these DC. Again this is likely to substantially influence the outcome. It will enrich the paper if these aspects are discussed.

3) The observation that vaccination with the MCA 205 TAP-deficient tumor protects non-specifically against RMA-S is not unexpected, because of the existence in RMAS and likely MCA 205 TAP deficient of so-called TEIPP antigens (T cell epitopes associated with impaired peptide processing), first described by Thorbald van Hall et al. in 2006 and lately last revisited in J. Exp. Med. 2018. These epitopes are generated from normal non-mutated household epitopes in a TAP-independent fashion, but being at low numbers, never make it to the cell surface in TAP-competent cells. However, in TAP-deficient cells they see their chance to load MHC class I in the absence of this competition.

4) The authors assume that the antigenic peptide is cleanly produced in the cells by cleavage of the ER translocation signal. Such cleavages are often influenced by surrounding sequences which in this case include the antigenic peptide. So it is possible that what are thought of as perfect antigenic peptides may not always be so. This should at least be discussed.

5) No thought seems to have been given to other ER trimming events such as those caused by the ERAP1, the ER aminopeptidase that is well known to trim antigenic peptides and influence presentation. Selective effects of ERAP1 trimming could change the outcome of actual peptides presented on the cell surface. Because the analysis carried out is of DNA sequences, rather than of the presented peptides discrepancies might be explained by differential processing of antigenic precursors. Again, this needs to be discussed.

6) The original model system was developed in RMA/S cells, but in later experiments included MCA cells that were rendered TAP2-deficient by targeted mutation. Is peptide presentation in two cell types comparable? Especially if there are differences in ERAP1 expression among the two cell types.

7) How the authors compute the relative frequency of minigenes in the library is key to the analysis but it is unclear how this was done. Is it possible that results could be skewed by the number of viral integration events or amplification errors?

---

## [Author Response]

Essential revisions:1) The authors have not reckoned with a potential presentation of epitopes by MHC class II and possible effects of CD4 help. It is well known that optimal CD8^+^ T cell immunity depends bot for effector cell induction and memory on specific CD4^+^ T cell immunity. Specific help is better than non-specific help, because the CD4^+^ T cells not only assist in the induction and programming of optimally activated CD8^+^ T cells, but also improve the intra-tumoral conditions for activity of CD8^+^ T cells. In the average neo-antigen carrying human tumors both CD4 and CD8 neo-epitopes exist (see Ott et al., 2017 and Sahin et al., 2017 papers quoted in the manuscript). It would be great to probe the rules for immunogenicity ageing in the presence or absence of specific helper epitopes being expressed and presented from the same RMA-S tumor, but I appreciate the fact that this is asking too much at the present time. Nevertheless it is quite possible that smaller fractions of tumor cells would become eliminated in the presence of robust and tumor-specific CD4^+^ help. The importance of CD4^+^ help has been discussed recently in Nature Reviews Cancer by Jannie Borst et al., 2018.

The lack of tumor-specific CD4 T cell help in our model system is a critical point and we agree with the comments from the reviewers that this should be discussed in the manuscript. We have added the following statement to the manuscript to address the question of CD4 T cell help and additionally expanded on our thoughts below:

“Our experimental design does not allow us to assess the role of CD4 T cells in CD8 T cell mediated anti-tumor responses. […] Moreover, there is the confounding possibility that some CD4 help is coming from unrelated antigens, such as mutated proteins found in RMA/S or the exogenous proteins introduced into the cells by the PresentER vector.”

The reviewers make the excellent point that the priming activities of helper CD4 T cells may increase the quality of some CD8 antigens. We strongly agree that CD4 cells likely play an important role in increasing the potency of some poorly immunogenic MHC I antigens. However, we think it is unlikely that CD4 help could normalize T cell responses such that all CD8 antigens would then have the same potency. In other words: even if CD4 cells can help some subclonal antigens to yield more potent immune responses, some antigens would still be more potent than others and this would be reflected in differential ability to clear subclonal cells.

Our ability to study MHC II ligands with our present method is limited (although we are considering building a CD4 version of PresentER). We have not worked out whether our signal sequence method efficiently yields MHC II peptides to APCs. In addition, the concept of pooled libraries of precisely defined MHC I antigens cannot yet be combined with libraries of longer MHC II peptides for technical reasons: two libraries would need to be made, one encoding MHC I peptides and the other encoding MHC II peptides, and there is no existing way to ensure that the quantity of the matched MHC I and II peptides are comparable across two libraries). Moreover, MHC II peptides are more difficult to predict computationally than MHC I peptides. We can imagine experiments with individual model antigens to test if the addition of a defined MHC II epitope to the tumor cells would improve T cell rejection of immunogenic subclones. However, a great deal of preliminary experiments would be needed to first show that MHC II peptides were being generated and seen in the RMA/S model we have defined and there are many caveats that would need to be addressed, e.g. the relative strengths of the CD8 and CD4 ligands, the possibility of CD4 help coming from an unrelated antigen, etc. In sum, we think this is a highly complex problem for which there are many experimental unknowns. We have not yet begun this work and we envisage it would take greater than 1 year to construct the system and to be confident in any assessment. As such, this is better suited for a follow-up manuscript.

2) Tumor heterogeneity may also involve the fact that some tumors are very rich in APC, while others are not, lacking the appropriate chemokine (receptor) expression. This type of heterogeneity might heavily influence the results one might obtain with the system used by the authors. Moreover most tumors lack intrinsic danger signals to activate these DC. Again this is likely to substantially influence the outcome. It will enrich the paper if these aspects are discussed.

We agree. This is also an excellent point. We have added to the Discussion the following statement:

“The heterogeneity in actual human tumors is characterized by differential infiltration with antigen presenting cells, which can change the sensitivity of the immune system to detect immunogenic antigens present at low clonal fractions, and thus could impact our findings. […] It would be very interesting to understand how different levels of antigen presenting cells contribute to dynamic thresholds of T cell immunogenicity.”

3) The observation that vaccination with the MCA 205 TAP-deficient tumor protects non-specifically against RMA-S is not unexpected, because of the existence in RMAS and likely MCA 205 TAP deficient of so-called TEIPP antigens (T cell epitopes associated with impaired peptide processing), first described by Thorbald van Hall et al. in 2006 and lately last revisited in J. Exp. Med. 2018. These epitopes are generated from normal non-mutated household epitopes in a TAP-independent fashion, but being at low numbers, never make it to the cell surface in TAP-competent cells. However, in TAP-deficient cells they see their chance to load MHC class I in the absence of this competition.

We agree that TEIPPs deserve a fuller discussion and thank the reviewers for pointing out that vaccination with MCA205 may be semi-protective against RMA/S tumors because of shared TEIPPs. Although we were aware of the existence of TEIPPs, we did not consider the possibility that shared TEIPPs could confer an antigen-specific protective effect. We have added this possible explanation, among others, to the Discussion as follows:

“The observation that animals prophylactically vaccinated with MCA205∆Tap2 cells are partially protected from outgrowth of RMA/S tumors may be due to a number of distinct possibilities. […] Although RMA/S and MCA205∆Tap2 are unlikely to share neoantigens derived from somatic mutations, they may share TEIPPs that may contribute to protection against RMA/S tumor growth.”

4) The authors assume that the antigenic peptide is cleanly produced in the cells by cleavage of the ER translocation signal. Such cleavages are often influenced by surrounding sequences which in this case include the antigenic peptide. So it is possible that what are thought of as perfect antigenic peptides may not always be so. This should at least be discussed.5) No thought seems to have been given to other ER trimming events such as those caused by the ERAP1, the ER aminopeptidase that is well known to trim antigenic peptides and influence presentation. Selective effects of ERAP1 trimming could change the outcome of actual peptides presented on the cell surface. Because the analysis carried out is of DNA sequences, rather than of the presented peptides discrepancies might be explained by differential processing of antigenic precursors. Again, this needs to be discussed.

Thank you for the points made in #4 and #5. We agree that the caveats and limitations of the PresentER system should be discussed more fully, especially as they apply to libraries of MHC-I antigens. Some of the considerations have been addressed in our first manuscript on the PresentER technology (see Gejman 2018 doi: https://doi.org/10.1101/267047). To be more explicit in the present paper, we have added a paragraph to the manuscript to discuss these points. Of note: we have not yet found evidence of alternative cleavage sites in any of the model antigens we have studied, however our tools to examine this phenomenon are limited (and would require to conduct MHC IP + mass spectrometry to show that cleavage/alteration is occurring). Nevertheless, we agree that some encoded peptides may cause alternative or no cleavage of the signal peptide leading to altered peptide or no free peptide.

The following paragraph is added:

“The PresentER system we have employed does have some potential biochemical caveats that may impact the interpretation of the data. […] Some peptides may lead to improper or no cleavage of the signal peptide whereas other peptides may be shortened (e.g. by ERAP1, an ER associated endopeptidase), altered or destroyed in the endoplasmic reticulum.“

6) The original model system was developed in RMA/S cells, but in later experiments included MCA cells that were rendered TAP2-deficient by targeted mutation. Is peptide presentation in two cell types comparable? Especially if there are differences in ERAP1 expression among the two cell types.

The reviewers are correct that differences in the antigen presentation machinery of MCA205∆Tap2 and RMA/S might affect the peptides presented in the library setting and we have added a sentence to the Discussion to address this:

“A caveat to the vaccination experiments is that differences between the antigen presentation machinery of MCA205∆Tap2 and RMA/S cells may lead to incomplete congruence in the PresentER-encoded MHC-I peptidome of these two cell lines. These differences may lead to some peptides encoded by the library to be poorly presented in one cell line but well presented in another cell line.”

In general, we believe that this caveat would only impact a small fraction of the members of the library. Moreover, we do not think that these data would change the interpretation of the results because MCA205∆Tap2 cells were only used to boost the initial T cell responses in the pooled library setting. Our later data suggests that there may be a fundamental roadblock to identifying immunogenic peptides in pooled library experiments, namely that low frequency antigens do not lead to effective T cell response.

7) How the authors compute the relative frequency of minigenes in the library is key to the analysis but it is unclear how this was done. Is it possible that results could be skewed by the number of viral integration events or amplification errors?

We agree. This is an excellent point. In general, pooled library experiments can be skewed by viral integration events or amplification errors. We have taken precautions at the level of PCR and viral transduction to avoid these issues. We have added several clarifying statements in the Materials and methods and provide more narrative detail here.

In order to avoid amplification bias due to high PCR cycle numbers or multiple integration events, PCR was performed with the minimum number of cycles necessary to see a clear band on an ethidium bromide gel (i.e. distinct from the gDNA smear). The minimum number of PCR cycles was empirically-determined for each batch of samples, but was typically 22 cycles for outer PCR and 18 cycles for inner PCR when the template was genomic DNA. These are relatively low cycle numbers that do not exhaust available dNTPs or primer (which can lead to amplification errors with >30 cycles). All steps were performed in PCR hoods or other clean environments with precautions to avoid cross contamination between samples. The maximum amount of genomic DNA that could be used in each reaction without inhibiting PCR was used in order to minimize amplification biases in each well. In order to avoid under-sampling low abundance minigenes, each sample was amplified using DNA equivalent to >1,000-fold the number of minigenes in the library. For instance, for the 5,000 minigene library samples used in this manuscript, at least 8e6 cell equivalents of gDNA (6µg of gDNA per cell => 48µg of gDNA) was amplified across >40 PCR reactions (we have subsequently optimized this using Q5 polymerase and can now load 10µg of gDNA per reaction). The total amount of gDNA from which to amplify was determined by examining H&E slides of RMA/S tumors and noting that growing tumors were comprised mainly of sheets of transformed cells, indicating that most gDNA comes from tumor/minigene-bearing cells and not stromal cells lacking the minigene construct. If we had noted more stromal infiltration, we would have amplified from a larger number of total cells (i.e. more total gDNA) in order to avoid under-sampling low abundance minigenes.

Biases arising from multiple viral integration events were avoided by transducing target cells at low multiplicities of infection (MOI). The functional titer was determined empirically for each batch of virus by aliquoting+freezing concentrated viral supernatant and transducing RMA/S (or MCA205) with a range of viral supernatant amounts. The amount of viral supernatant corresponding to <⅓ maximal transduction efficiency was used to generate the library, suggesting that on average <10% of cells were doubly infected. >1,000 more cells than minigenes in the library were transduced (usually 2-3,000) and maintained at all times (including during puromycin selection) to ensure that low abundance minigenes did not drop out of the pool.

The quantification of minigenes in the library after an experiment was performed as follows: genomic DNA was extracted from tumors, the minigenes were amplified by nested PCR and sequenced on Illumina instruments. After sequencing, all reads were aligned to the minigenes in the library. The number of reads aligning to each minigene was divided by the total number of reads aligning to the library to obtain the relative abundance of each minigene.